# Study of the Plastic Behavior of Rough Bearing Surfaces Using a Half-Space Contact Model and the Fatigue Life Estimation According to the Fatemi–Socie Model

**Flavien Foko Foko \*, Lukas Rüth \*** , **Oliver Koch** **and Bernd Sauer**

Chair of Machine Elements, Gears and Tribology (MEGT), University of Kaiserslautern Landau (RPTU), 67663 Kaiserslautern, Germany
\* Correspondence: fokofoko@mv.uni-kl.de (F.F.F.); lukas.rueth@mv.uni-kl.de (L.R.)

**Abstract:** A multiscale approach for the fatigue life estimation of rolling bearings is presented in this paper and applied to inner rings of cylindrical roller bearings of the type NU208. The forces acting in the rolling contact are determined from system-oriented modeling at the macro level. The microscale contact simulations are carried out in a half-space contact model. The stresses on the inner ring are determined and used in the local fatigue approach, according to Fatemi–Socie, for fatigue life estimation. Four surface variants were investigated, one ideally smooth surface and three real (rough) surfaces. The three rough surface variants used different finishing processes: fine ground, hard turned, and rough ground were produced. A load case with a maximum pressure of 2.4 GPa in the roller-inner ring contact was investigated. In addition to the fatigue life estimation, the plasticity behavior (surface topography, resulting contact pressure, and residual stress) of the three manufactured surfaces stood in the focus of the work. As the comparison between experimental and simulated results confirms, good predictions can be made with the simulation model.

**Keywords:** fatigue life; fatemi–socie; half space; rolling bearing; rolling fatigue

## 1. Introduction

The current fatigue life calculation of rolling bearings according to ISO/TS 16281 [1] requires raceway surfaces with high quality. To meet the required surface quality, special manufacturing processes such as honing are used, which are associated with high costs. The use of other processes for the more cost-effective production of rolling bearing inner rings should first be preceded by an analysis of the tribological behavior of the resulting surface. In [1], several formulas with different levels of detail are described. The nominal $L_{10}$ for bearings under "usual conditions" can be determined as well as the modified reference service life $L_{10mr}$ by adding the fatigue life coefficient also. However, the influences of plasticity and the local surface roughness value are not considered. Based on the work of Brown and Millers [2], a fatigue approach, in which the material parameters and the stress history are considered, was proposed by Fatemi and Socie. This model is used in this work and presents a local approach that allows a realistic estimation of the fatigue life by evaluating the stresses on a local volume element.

To date, the Fatemi–Socie model has been used to simulate the highly loaded rolling contact, e.g., in [3]. An essential aspect that has been ignored until now is the consideration of the influence of plasticity in the model. As a novelty, in this work, besides the consideration of the real surface topography, an approach to consider the plasticity influence in the Fatemi–Socie model is presented. A central question remains, the procurement of the required cyclic material parameters for the determination of the local stressability. For this purpose, the results of the FKM (Research Association for Mechanical Engineering) method from [4,5] proposed by the Institute for Mechanical Plant Engineering and Structural Durability (IMAB) of the Clausthal University of Technology are used.

Finite element (FE) models or, as in this work, semianalytical contact models based on half-space theory can be used to simulatively investigate the plastic behavior of the rough surfaces in contact. The latter offers the possibility of solving high-resolution contact problems on the microscale with a reasonable computation time compared to the FE model. The implemented semianalytical contact model is based on the contact algorithm of Polonsky and Keer [6,7] for normal contact with elastic material behavior and the extension according to Jacq [8,9] to consider plastic material processes.

In addition to contact modeling, the realistic definition of the simulation parameters is of major importance. These include the geometry, material, and load parameters. The real surface of the inner rings (IR) is determined by means of confocal measurement and integrated into the model, whereas the counter surface of the cylindrical roller is assumed to be ideally smooth. The Swift hardening model [10] is used to describe the plastic material's behavior.

The macro load in the rolling contact depends on many system parameters during operation and is determined in the single-bearing model of a multibody simulation (MBS) model. From the macro load, the load defined in the half-space contact model on the microscale is derived.

The focus of this work is to develop a multiscale approach to life estimation of cylindrical roller bearings considering the plasticity of real contact surfaces.

## 2. Materials and Methods

In this section, the entire modeling approach is presented as an introduction. In Section 2.1, the contact models are briefly introduced. The focus will be on the parameters of the macromodel and the resulting parameter values for the microcontact model. The second Section 2.2, will present a brief comparison between the elastic solution in the half-space model and the Hertzian theory. This comparison will serve to validate the elastic stresses calculated in the half-space model, as they are important input variables for Jacq's half-space plasticity model used later. To solve the plasticity problem, one also needs a model that describes the flow behavior of the material. Here, the Swift hardening model given in Equation (1) is used.

The modeling strategy is based on two simulation levels as schematically shown in Table 1.

**Table 1.** Geometric and load variables of the MBS model and half-space model.

| Macro-Variables | | Micro-Variables | |
|---|---|---|---|
| $D_R/\text{mm}$ | 10 | $L_x/\text{mm}$ | 0.6 |
| $D_{IR}/\text{mm}$ | 49.98 | $L_y/\text{mm}$ | 0.6 |
| $L_{eff}/\text{mm}$ | 9.7 | | |
| $F_{macro}/\text{N}$ | 4986 | $F_{micro}/\text{N}$ | 310 |

In the first level, the load situation is determined using an individual bearing model built with the tool LaMBDA (which stands for bearing multibody calculation and dynamics analysis) [11]. LaMBDA is not a stand-alone tool, but a plug-in for the commercial MBS software SIMPACK. LaMBDA essentially consists of a user-friendly interface for model generation and the necessary high-quality contact models. Here, various influencing variables (e.g., rolling element and raceway profiling) are taken into account by means of an in-house developed calculation routines [11–13].

In the half-space model, the individual contact between the rolling element and the inner ring volume area $V_{IR}$ below it is solved on the microscale. Here, individual static contact analyses are carried out and thus the maximum contact force in the contact RE/IR over the angular position is considered. The force curve shown in Figure 1 (dashed line) represents a simplification. In reality, a volume element at the IR undergoes brief load peaks (peaks marked with * in Figure 1 top right) during the passage of the load zone. The

simplification used is therefore a worst-case estimation. For each angle-position-dependent normal force, the equivalent stress, $\sigma_v$ according to Tresca, is calculated in each volume element of the previously discretized calculation area. The volume element with the maximum equivalent stress at the highest load $F_{max}$ (at the angular position $\psi_{Fmax}$) is defined as the critical volume element. The angular-position-dependent component of the stress tensor at the critical volume element is finally used in the Fatemi–Socie multiaxial fatigue criterion to determine a critical number of load cycles $N_f$.

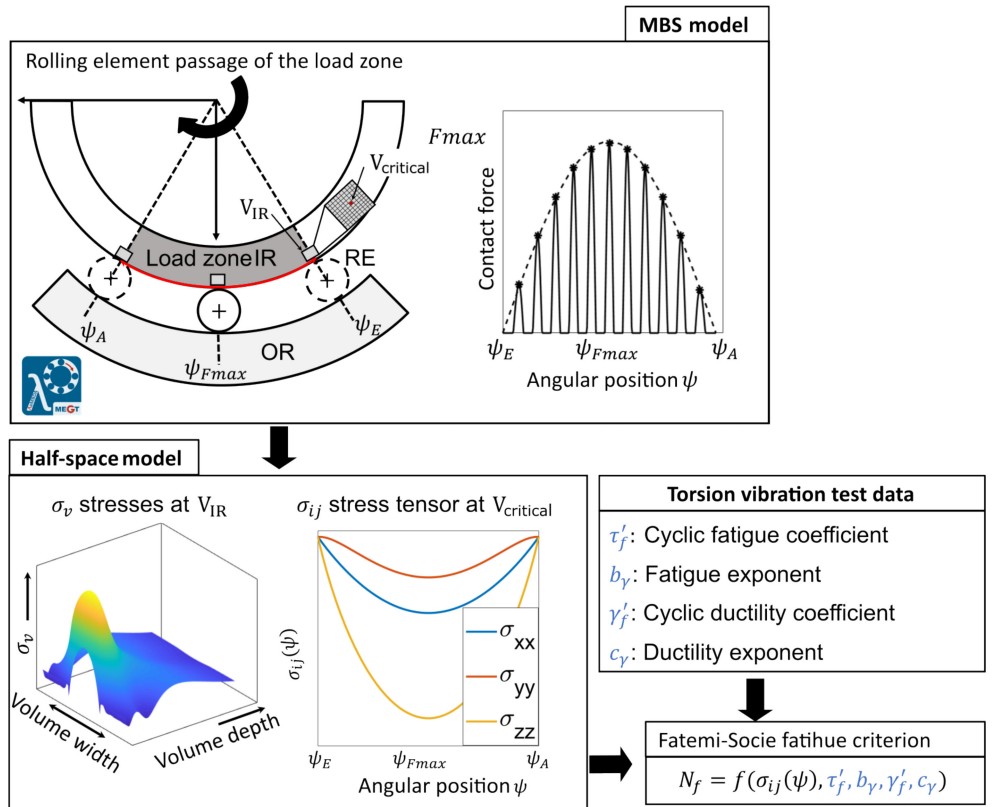

**Figure 1.** Schematic illustration of the entire modeling approach based on two contact models: MBS model in which the rolling element loads are determined from the entry into the load zone (at the angular position $\psi_E$ ) to the exit (at the angular position $\psi_A$), half-space model in which the stresses resulting in the inner ring (IR) (X: axial direction, Y: circumferential direction, Z: depth direction) are calculated and serve as input for the fatigue life estimation in the Fatemi-Socie model.

### 2.1. Contact Models

The maximum rolling bearing load is determined in the MBS model for the investigated load case. This load results in the maximum pressure of 2.4 GPa in the half-space contact model in the case of an ideally smooth inner ring surface. Figure 2 illustrates the contact between the rolling element and inner ring; (a) in the MBS model and (b) reduced in the half-space model. The half-space model is characterized by a definition of the geometry and load on the microscale. Table 1 summarizes the geometry and the load parameters.

The roller diameter $D_R$ and inner raceway diameter $D_{IR}$ given here as macroscopic values are valid for a cylindrical roller bearing of the type NU208 and were measured. In reality, the roller does not have an ideal cylindrical shape but is crowned and therefore does not carry over its entire length. This results in the effective contact length $L_{eff}$. The macroforce is the maximum value of the roller inner ring contact force (see Figure 2). computed in the MSB model.

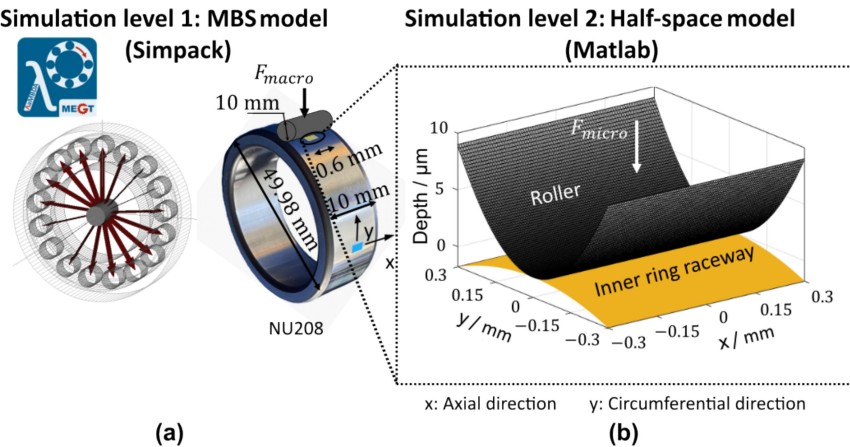

**Figure 2.** Illustration of the contact models used in both simulation levels, (**a**) macroscale MBS model and (**b**) microscale half-space model.

For the definition of the calculation area, it must be ensured that the maximum theoretical contact zone is covered. The Hertzian solution is used to determine the theoretical contact width. For the macrocontact defined by the parameters given in Table 1, a half contact width $a_{\mathrm{H}}$ of approx. 1.5 mm was calculated according to Hertz. The lengths of the calculation area for the microcontact $L_{\mathrm{x}}$ and $L_{\mathrm{y}}$ were each defined as four times $a_{\mathrm{H}}$. For this calculation area with the length $L_{\mathrm{x}}$ and $L_{\mathrm{y}}$, we determined the microlevel of the force $F_{\mathrm{micro}}$ given in Table 1. For this purpose, the macrocontact force $F_{\mathrm{macro}}$ acting on the single contact in the MBS model is scaled down with the length ratio $L_{\mathrm{y}}/L_{\mathrm{eff}}$.

Figure 3 shows the maximum contact forces between the rolling element and the inner ring determined for the investigated load case in the MBS model.

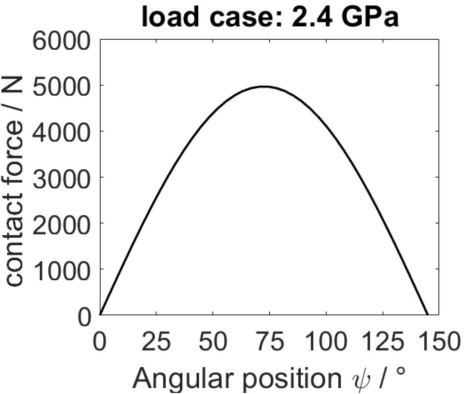

**Figure 3.** Maximum contact forces in the rolling bearing inner ring contact were determined using the MBS model.

### 2.2. Comparison Half-Space vs Hertz Theory

For validation purposes of the elastic stresses calculated within the half-space model, Figure 4 shows a comparison with results from Hertz's theory. A good match between the two results is found with regard to the stress component (a) and the von Mises equivalent stress (b) calculated from it. The maximum equivalent stress is at a depth of 0.124 mm. From the graph of the von Mises equivalent stress, no plastic deformation is to be expected for a microyield stress of 1477 MPa according to [14] in the simulation case with ideally smooth inner ring geometry.

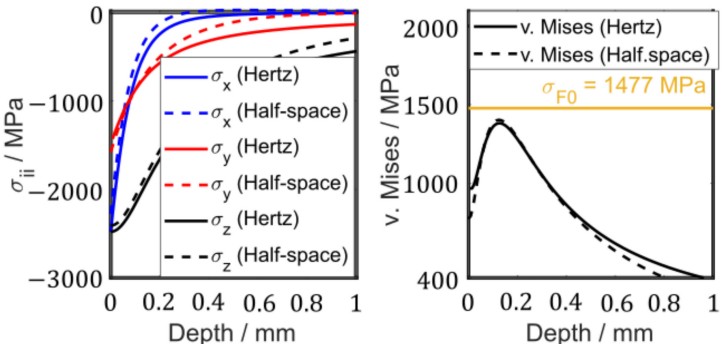

**Figure 4.** Comparison of the half-space elastic contact results with those of Hertz theory using the example of a line contact in a cylindrical roller bearing NU208 for the load case 2.4 GPa for an ideally smooth inner ring geometry.

For simulation cases with real surfaces, the roughness results in locally high contact pressures that lead to plastic deformation. In this work, the 3D half-space plasticity model presented in detail by Jacq in [8,9] is used to determine the resulting plastic surface deformation and residual stresses. In this model, a yield function is required to describe the material hardening during loading. The Swift strain hardening model [15] is described by the following expression:

$$\sigma_F = B\left(C + \varepsilon_p\right)^n, \tag{1}$$

was chosen and the material values given in Table 2 were used.

**Table 2.** Microplastic material parameters of the Swift hardening model for 100Cr6 rolling bearing steel [10,14,16,17].

| Parameters | Value |
|:---:|:---:|
| $B$/MPa | 945 |
| $C$/- | 40 |
| $n$/- | 0.121 |

The use of constant plastic material behavior for the entire material depth as done in this work is a simplification. Works such as [18] could show the depth dependence (called the indentation size effect) of the hardness and thus also of the plastic material parameters.

### 3. Fatemi–Socie Fatigue Life Approach

The Fatemi–Socie model comes from fracture mechanics and is based on the observation that for many metals, crack initiation occurs at a critical volume element in the critical plane with maximum shear strain, and the normal stress perpendicular to this plane accelerates the fatigue process by crack opening (see Figure 5)

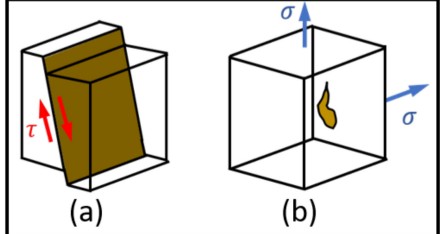

**Figure 5.** Schematic representation of the two basic concepts of material fatigue considered in the Fatemi–Socie model, (**a**) initiation of the slip bands (also places of microcrack initiation) by the shear stress τ, (**b**) microcrack expansion by the normal stress σ.

The following relationship has been proposed [19]:

$$\gamma_{a,eq} = \gamma_{max}\left(1 + k_e \cdot \frac{\sigma_{n,max}}{\sigma_y}\right) = \frac{\tau'_f}{G}\left(2 \cdot N_f\right)^{b_y} + \gamma'_f\left(2 \cdot N_f\right)^{c_y}. \tag{2}$$

The left-hand side of Equation (2), often called the damage parameter DP [20], describes the load by an equivalent shear strain amplitude $\gamma_{a,eq}$. The right-hand side, on the other hand, indicates the material strength as a function of material parameters from cyclic torsion pendulum tests.

The uniaxial cyclic material parameters can be found for most rolling bearing steels in the standard literature, e.g., [21]. A current method (FKM) for estimating cyclic material properties for the material groups steel, cast steel, and aluminum alloys is proposed in [4,5]. The FKM method is based on a large database with quasistatic and strain-controlled test results, and only requires the tensile strength $R_m$ as an input value. The results of the FKM method are summarized in Table 3

**Table 3.** Uniaxial cyclic material properties for the material group steel according to the FKM method [4,5] as a function of tensile strength.

| Parameters | Value |
| :---: | :---: |
| $\sigma'_f$ | $3.1148\left(\frac{R_m}{MPa}\right)^{0.897}$ |
| $b$ | $-0.097$ |
| $\varepsilon'_f$ | $\min\begin{pmatrix} 0.338 \\ 1033\left(\frac{R_m}{MPa}\right)^{-1.235} \end{pmatrix}$ |
| $c$ | $-0.52$ |
| Validity range | $R_m = 121\ldots2296$ MPa |

The required multiaxial cyclic material parameters ($\tau'_f$, $b_y$ and $c_y$) in Equation (2) can be derived from the uniaxial quantities in Table 3 as described in [22] using the following relationships:

$$\begin{cases} \tau'_f \approx \frac{\sigma'_f}{\sqrt{3}} \\ \gamma'_f \approx \sqrt{3}\varepsilon'_f \\ b_y \approx b \\ c_y \approx c \end{cases}. \tag{3}$$

Table 4 summarizes the values used in the fatigue life calculation in the Fatemi–Socie model for 100Cr6 (at 60 HRC and Rm = 2016 MPa):

**Table 4.** Material parameters using in the Fatemi–Socie fatigue life model.

| Parameters | Value |
| :---: | :---: |
| $\sigma_y/$MPa | 1953 |
| $\tau'_f/$MPa | 1656 |
| $G/$MPa | 80,769 |
| $\gamma'_f/-$ | 0.148 |
| $b_y/-$ | $-0.097$ |
| $c_y/-$ | $-0.52$ |

### 3.1. Determination of the Damage Parameter in the Fatemi–Socie Model

The equivalent shear strain amplitude in the Fatemi–Socie model, also called the damage parameter, describes the load on the volume element by two quantities, the maximum shear strain amplitude $\gamma_{max}$ and maximum normal stress $\sigma_{Nmax}$ perpendicular to the critical plane. The determination of the maximum shear strain amplitude will be

discussed first, followed by a method for considering the normal stresses in the Fatemi–Socie model for the general rolling contact problem.

The starting values are the stress components $\sigma_{ij}$ to the respective angular positions at the selected critical volume element, as shown in Figure 6.

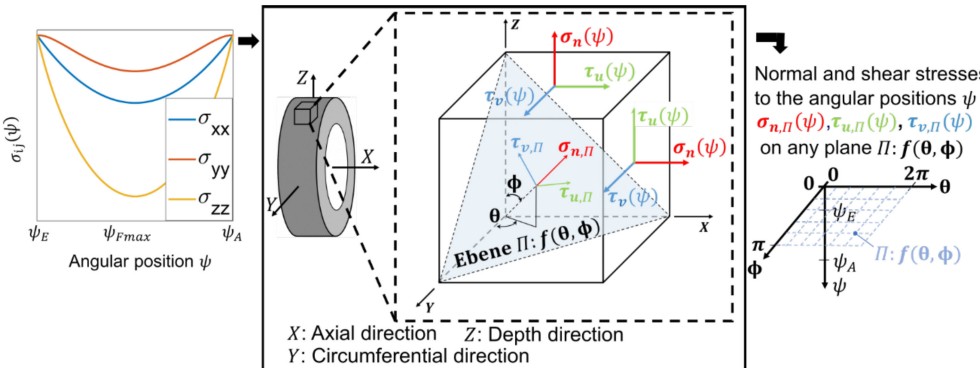

**Figure 6.** Left: stress component on a critical volume element for each angular position, middle: determination for each angular position of the normal and shear stresses on any plane $\Pi$ of the critical volume element, right: result as input variables for the determination of the shear stress vectors.

The volume element with the maximum shear stress is selected as the critical volume element for the rolling bearing contact. To determine the critical plane, an angle discretization $f(\theta, \phi)$ ($\theta = 0 : \cdots 2\pi$; $\phi = 0 : \cdots \pi$) and the relations [23]:

$$\vec{n_\Pi} = \begin{pmatrix} \sin\theta\cos\phi \\ \sin\theta\sin\phi \\ \cos\theta \end{pmatrix}; \; \vec{v_\Pi} = \begin{pmatrix} \cos\theta\cos\phi \\ \cos\theta\sin\phi \\ -\sin\theta \end{pmatrix}; \; \vec{u_\Pi} = \begin{pmatrix} -\sin\phi \\ \cos\phi \\ 0 \end{pmatrix}, \quad (4)$$

are used to determine the normal vector $\vec{n_\Pi}$ and the tangential vectors $\vec{v_\Pi}$, $\vec{u_\Pi}$ to any plane $\Pi$ at the critical volume element. Using these vectors, the normal stress $\sigma_{n,\Pi}(\psi)$ and shear stresses $\tau_{u,\Pi}(\psi)$, $\tau_{v,\Pi}(\psi)$ on the respective plane $\Pi$ are determined. For each plane, the angular position dependent shear stress vectors are plotted as illustrated in Figure 7, and the radius of the smallest enclosing circle is determined as the maximum shear stress amplitude $\Delta\tau_{max}/2$.

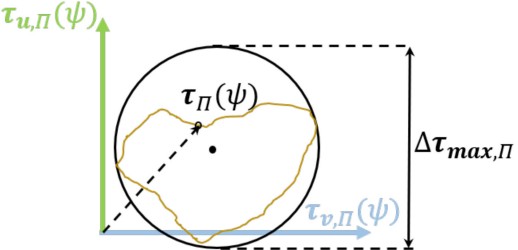

**Figure 7.** Plotting the angular position dependent shear stress vectors for a plane $\Pi$ to determine the shear stress amplitude.

The critical plane $\Pi$ is chosen as the one with the largest maximum shear strain amplitude. The searched maximum shear strain amplitude $\gamma_{max}$ in Equation (2) is determined from the maximum shear stress amplitude $\Delta\tau_{max,\Pi}/2$ and the shear modulus $G$ [3]:

$$\gamma_{max} = \frac{\Delta\tau_{max,\Pi}}{2G}. \quad (5)$$

### 3.2. Maximum Normal Stress in the Fatemi–Socie Model for the Rolling Contact Problem

In contrast to the maximum shear strain amplitude, there is ambiguity in the literature [3] about the determination of the maximum normal stress $\sigma_{N,max}$ in the Fatemi–Socie model in Equation (2). In the general complex multiaxial loading case, the maximum normal stress perpendicular to the critical plane is used in Equation (2). For rolling contact problems, as confirmed by the results of work [3], the maximum compressive normal stress $\sigma_{DN,\,max}$ (by magnitude with negative sign) on the critical plane should be used as the normal stress. This is based on the fact that the sensitivity of the material to the occurrence of cracks is reduced by the existing residual compressive stresses [24]. Compared to the linear elastic calculation, the consideration of plasticity leads to a reduction of the roughness by flattening and an increase of the true contact area. A higher true contact area means a lower maximum contact pressure value and lower normal stresses.

## 4. Experimental and Surface Topography Measurement Setup

The following section gives an overview of the manufacturing of the specimens (Section 4.1), the test setup for the experimental investigation of the fatigue life (Section 4.2), and the surface measurements on the manufactured as well as the tested bearing rings.

### 4.1. Specimen Manufacturing

Specially manufactured inner rings for the rolling bearing tests with three different finishing processes that produced three different surface conditions were used, namely fine ground, rough ground, and hard turned. It should be noted that the fine ground condition is the current state of the art and is achieved through grinding and subsequent honing. All rings were made of 100Cr6 (AISI 52100), with a chemical composition in accordance with DIN EN ISO 683-17 [25]. The manufactured new inner rings replaced the original ones for the fatigue tests. The surfaces resulting from the manufacturing process are shown in Section 4.3.

### 4.2. Bearing Test Rig

A rolling bearing life test rig, as shown in Figure 8, was used to conduct the tests. The rig allows for the simultaneous testing of four bearings under comparable loading conditions. The radial force, $F_{rad}$, was applied using an actuator located on a hollow shaft that contains two bearings and interposed disk springs. These inner bearings transfer the force to the shaft, which is connected to the two outer bearings. Due to the symmetrical arrangement of the outer and inner bearings, all bearings experienced equal loading. A circulating lubrication system was used for oil feed. To detect fatigue damage at an early stage during the tests, a vibration-based condition monitoring system was utilized [26].

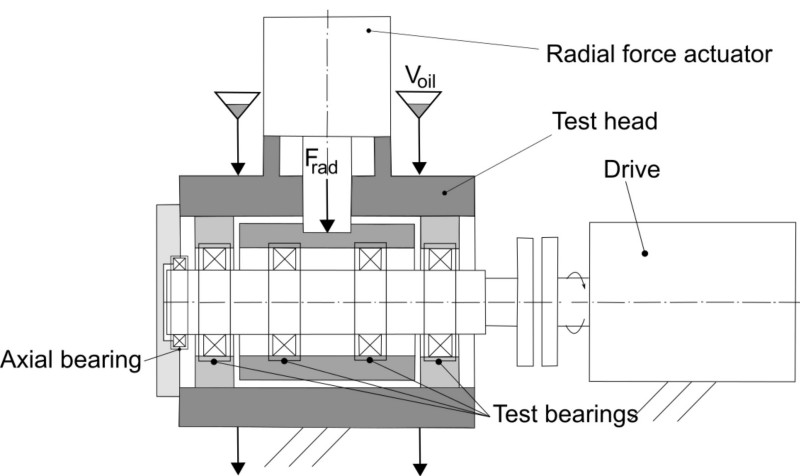

**Figure 8.** Schematic illustration of the bearing test rig.

*4.3. Surface Measurement*

The surface topography was measured by means of a confocal microscope of the type μsurf of the company NANOFOCUS, Oberhausen, Germany (see data in Table 5). The so called Nipkow-disk is used to measure an area instead of a point. By adjusting the distance to the surface, individual focus images are generated. These single images are then combined to form a complete parametric 3D surface. The vertical traverse path needs to be sufficiently large to encompass the area from the lowest to the highest point of the surface that needs to be measured [27].

**Table 5.** Relevant characteristics of the 3D confocal microscope.

| Parameters | Unit | Value |
|---|---|---|
| Magnification | - | 20 |
| Measuring field | μm | $800 \times 800$ |
| Numerical aperture | - | 0.6 |
| Working distance | mm | 0.9 |
| Resolution in depth direction | nm | 4 |
| Resolution in the plane | μm | 1.6 |

The characteristic topography values were determined in accordance with DIN 25178-2 [28] using the software MountainsMap® 6.0 of the supplier DIGITAL SURF, Besançon, France. In this process, the raceway curvature and the positional deviations due to tilting are removed. This results in a topography projected onto a plane. In addition, the images are filtered by means of Gaussian filters, and the defects of individual pixels are corrected via interpolation.

The specially designed measuring device with two centering pins positioned in two centering holes (colored green in Figure 9) ensures that measurements are taken at the same point in the manufactured and the in the run-in state.

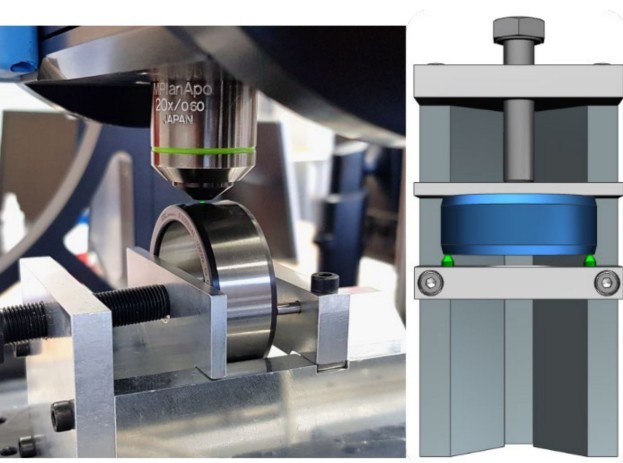

**Figure 9.** Surface analysis using a NANOFOCUS μSurf EXPLORER confocal microscope: Measurement setup (left) and CAD model (right).

## 5. Simulations Results

The results presented here are summarized in three sections. In Section 5.1, the tribological behavior of the measured surfaces in the manufactured and in the run-in states (after $10^7$ revolutions) is investigated. For this purpose, elastic contact simulations are carried out on the micro level with the help of the half-space contact model. The focus is on the change of the true contact area and the theoretical plastic zone for both states (manufactured and run-in states after $10^7$ revolutions). The theoretical plastic zone is defined as the set of all volume elements in the discretized calculation volume with a von Mises equivalent stress value higher than the yield stress of 1477 MPa.

Many effects contribute to the surface change during the operation. In this work, a surface change predominantly caused by plasticity is assumed and in the second Section 5.2, the plasticity of the surfaces and the material depth is investigated with the help of a 3D plasticity model. To validate the results obtained with the 3D plasticity model, the results of the elastic contact simulation with the real surface in the run-in state in Section 5.1 are used as a reference. The use of the real surfaces in the run-in state assumes that after $10^7$ revolutions all relevant flattening surface effects are completed.

In the last Section 5.3, fatigue life estimation is carried out according to the Fatemi–Socie model, considering the simulated elasto-plastic deformed surfaces in Section 5.2.

*5.1. Analysis of the Tribological Behaviour of the Measured Surfaces in Manufactured and In Run-In (after $10^7$ Revolutions) State by Means of Elastic Contact Simulation*

In order to analyze the stresses in the contact for differently finished bearing rings, elastic contact simulations were carried out. For this purpose, the confocally measured surface of the inner ring was brought into contact with an ideally smooth rolling element surface in the simulation model. The respective surface topographies of the bearing rings in the two studied cases are shown in Figure 10. For illustrating the roughness profile, an additional 2D section is shown in the right column of the figure.

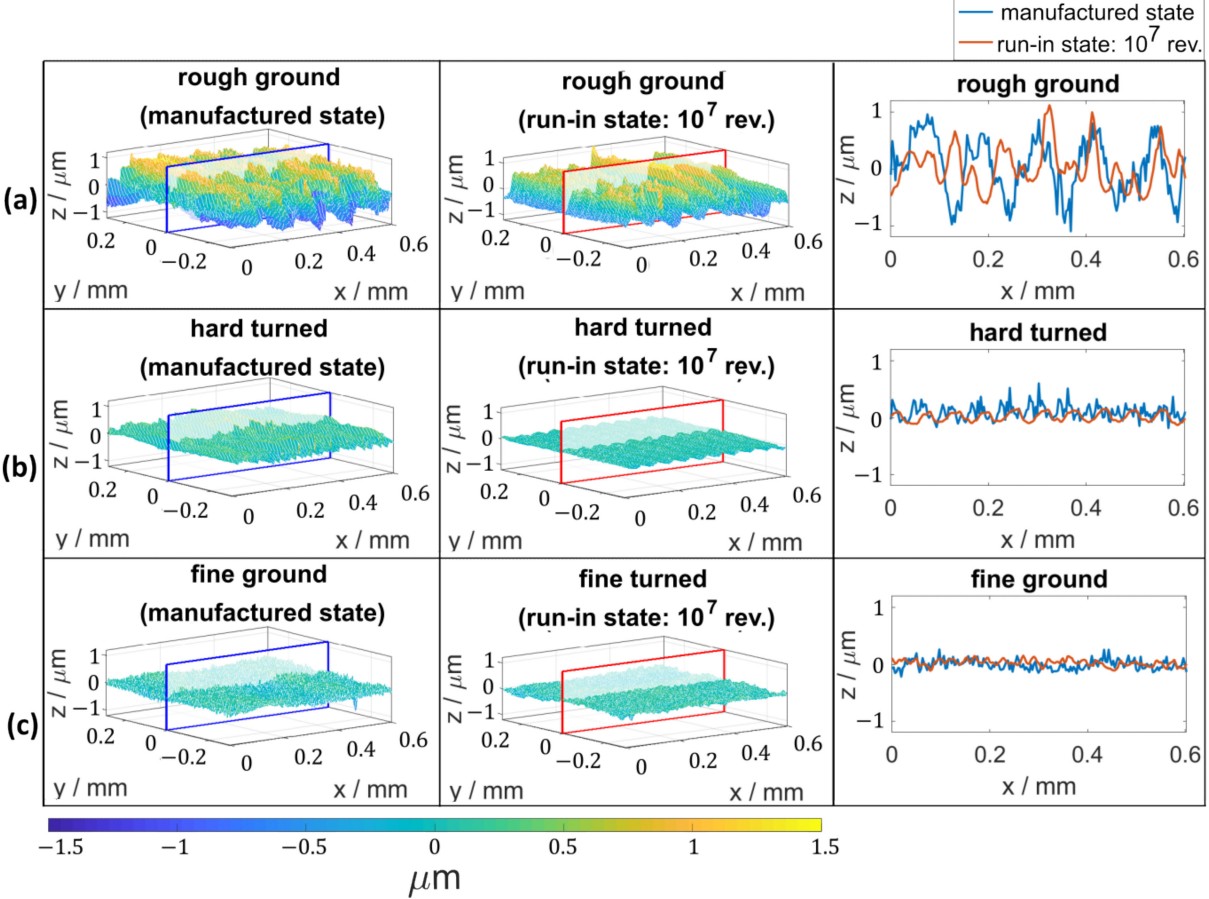

**Figure 10.** Measured real inner ring surface (**a**) rough ground, (**b**) hard turned, and (**c**) fine ground in manufactured and in run-in states.

To obtain a statement about the measurement error in the form of measurement noise, two repeat measurements were carried out for the identical measurement point in each case. Using the analysis software MountainsMap, common surface parameters such as the mean arithmetic height $S_A$ and the reduced peak height $S_{pk}$ were calculated as surface function parameters of the Abbot–Firestone curve, and the standard deviation was determined (see

Table 6). It should be noted that the section of $0.6 \times 0.6$ mm considered represents an exemplary section of the entire raceway surface.

**Table 6.** Surface-related roughness parameters $S_A$ and $S_{pk}$ in the manufactured state and in the run-in state, as mean value $\pm$ standard deviation of the repeat measurements.

| | Manufactured State | | Run-In State ($10^7$. rev.) | |
|---|---|---|---|---|
| | $S_A / \mu m$ | $S_{pk} / \mu m$ | $S_A / \mu m$ | $S_{pk} / \mu m$ |
| rough ground | $0.413 \pm 0.001$ | $0.392 \pm 0.003$ | $0.329 \pm 0.001$ | $0.310 \pm 0.001$ |
| hard turned | $0.168 \pm 0.002$ | $0.163 \pm 0.004$ | $0.113 \pm 0.003$ | $0.091 \pm 0.000$ |
| fine ground | $0.078 \pm 0.001$ | $0.061 \pm 0.002$ | $0.092 \pm 0.002$ | $0.083 \pm 0.002$ |

The data depicted in Figure 10 indicates that the rough ground variant exhibits a significantly higher roughness in its manufactured state than the other variants. Conversely, the fine ground variant demonstrates the lowest roughness peaks, as expected. Notably, both the hard turned and rough ground variants exhibit a significant reduction in roughness peaks compared to the fine ground variant. The difference in surface roughness between the hard turned and rough ground variants is particularly pronounced, implying that the flattening effect is more substantial in the hard-turned variant. The range of the standard deviation from 0 nm to 4 nm indicates very stable measurement results for the repeat measurements. The deviations may be caused by measurement noise, see for example [29–31].

A calculation area with the dimensions 0.6 mm, 0.6 mm, and 50 μm was defined in each case, which was discretized into 194, 194, and 129 elements in the axial (x), circumferential (y), and depth (z) directions. The applied micro force is 310 N and corresponds to a maximum contact pressure of 2.4 GPa in the case of an ideally smooth inner ring surface.

In order to compare the surface topography qualitatively, the true contact areas are evaluated, as indicated in Table 7. The larger the true contact area $A_{true}$, the more equal the load distribution, and the locally resulting stresses become smaller. In a manufactured state, the fine-ground and hard-turned surfaces have similar true contact areas of 0.1275 mm$^2$ and 0.1265 mm$^2$, while the rough-ground surface has the smallest true contact area of 0.0883 mm$^2$ and thus indicates a worst tribological behavior. After $10^7$ revolutions, the results of the contact simulations show an increase in the true contact area to 23%, 61%, and 49% for the rough-ground, hard-turned and fine-ground surfaces, respectively, which is due to the flattening of the roughness peaks. This leads to a slightly higher $A_{true}$ for the hard-turned variant compared to the fine-ground variant.

**Table 7.** Simulated true contact areas of the measured rough surfaces in manufactured and run-in (after $10^7$ revolutions) states.

| | Manufactured State | Run-In State ($10^7$ rev.) |
|---|---|---|
| | $A_{true} / mm^2$ | $A_{true} / mm^2$ |
| rough ground | 0.0883 | 0.1088 |
| hard turned | 0.1265 | 0.2037 |
| fine ground | 0.1275 | 0.1896 |

The purely elastically simulated contact pressures are shown in Figure 11. In the manufactured state, maximum pressure values of 15.9 GPa, 15.9 GPa, and 17.4 GPa are obtained for the rough ground, hard turned, and fine ground surfaces respectively. These pressure values result in the selected measuring area at the highest asperites of the respective surfaces. General conclusions cannot be formed on the basis of the sample measurement shown.

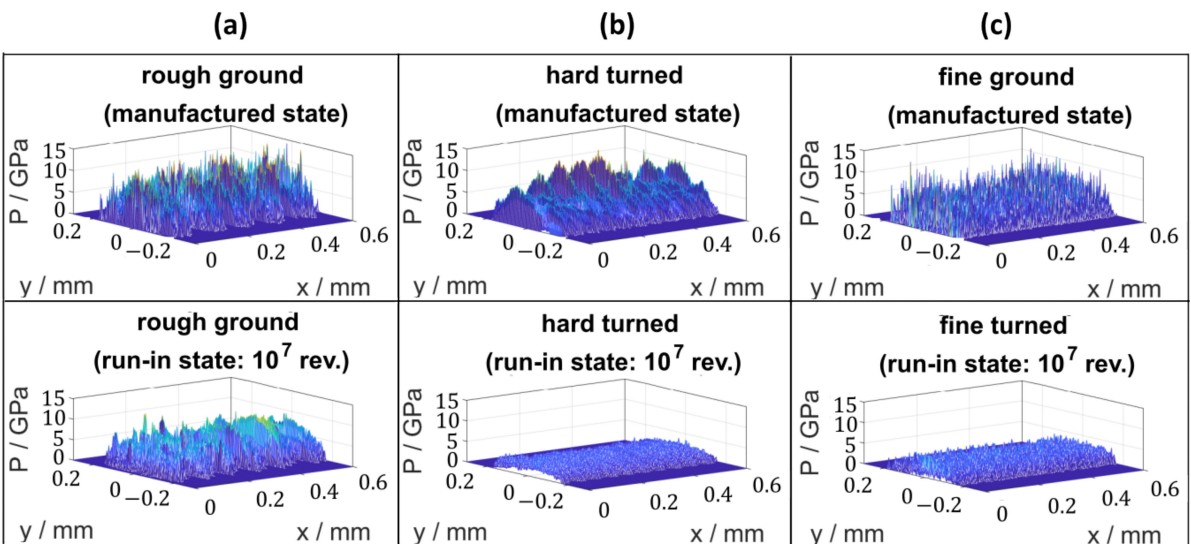

**Figure 11.** Elastically simulated contact pressures with of real surfaces (**a**): rough ground, (**b**) hard turned, and (**c**) fine ground bearing ring in manufactured (top) and run-in states (bottom).

The better topography (characterized by the largest true contact area) of the hard-turned surface is shown to be advantageous in the operation by the significant reduction of the maximum contact pressure from 15.9 GPa to 4.3 GPa in the calculation case with the real surface in the run-in state. With the fine-ground surface, there is also a large reduction in the maximum contact pressure from 17.4 GPa to 5.7 GPa. The smallest reduction from 15.9 GPa to 12.3 GPa is noted for the rough ground surface.

The results of the stress calculation for the differently finished bearing rings in the manufactured and run-in states are shown in Figure 12. By comparing the different surface variations, the rough ground surface shows the highest stresses in both states. These are located in a surface-near area. In the simulation case with surfaces in the run-in state, it can be observed that the stress states for the hard-turned and the fine-ground variant are in the same range.

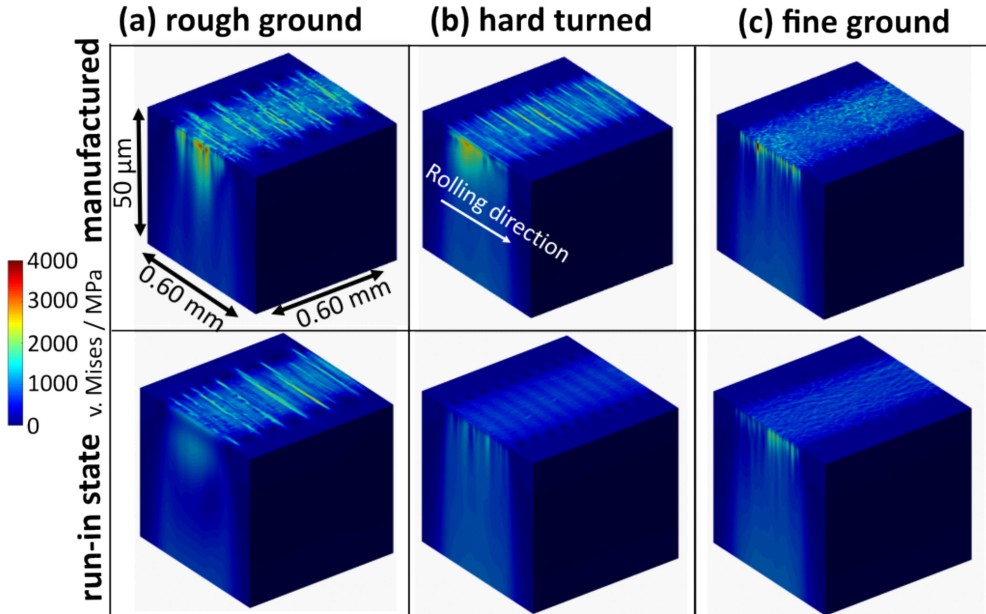

**Figure 12.** Elastically simulated vs. Mises stresses in contact of real surfaces; (**a**) rough ground, (**b**) hard ground, and (**c**) fine ground bearing ring in manufactured (top) and in run-in states (bottom).

Further analyses of the observed different stress distributions were carried out. To estimate the potential plastic zone, the volume elements with stresses above the equivalent yield stress of 1477 MPa were determined as given in Table 8.

**Table 8.** Calculated potential plastic zone as the number of volume elements $N_{vol}$ with a von Mises equivalent stress value greater than the yield stress, for the measured surfaces in manufactured and in run-in state (after $10^7$ revolutions).

| | Manufactured State | | Run-In State ($10^7$. Rev.) | |
|---|---|---|---|---|
| | $N_{vol}$ | Vol / % | $N_{vol}$ | Vol / % |
| rough ground | 434,817 | 8.95 | 433,501 | 8.92 |
| hard turned | 81,493 | 1.68 | 477 | 0.0098 |
| fine ground | 46,515 | 0.9581 | 491 | 0.0101 |

In the manufactured state, the rough-ground and fine-ground surfaces each have the largest and smallest theoretical plastic zone. However, the smallest theoretical plastic zone for the surface variant in run-in state results from the hard-turned surface. This result correlates with those of the true contact surface, which predicts the smallest stresses for the hard-turned variant.

*5.2. Simulation of the Plastic Behavior in the Surface-Near Area*

The results in Figure 12 show that plasticity is to be expected in the area close to the surface. For the simulative investigation of the plastic behavior of the inner rings, a cyclic quasistatic contact between the rolling element and the inner ring at the maximum load of 310 N (see Table 1) is simulated with the half-space model. The true contact area $A_{real}$, the contact pressures and the residual stresses are evaluated and compared with the results determined by means of an elastic contact simulation from the real measured surfaces in the run-in state after $10^7$ revolutions. The measured real surfaces in the manufactured state (see Figure 10 in the first column) are used as the input geometry of the contact model.

5.2.1. Predicting Surface Change Using Elasto-Plastic Contact Simulation

Figure 13a shows the results of the elasto-plastic simulated variation of the true contact areas. The 200 load cycles were simulated up to convergence. In the converged state, a true contact area, $A_{real}$, of 0.1260 mm$^2$, 0.1543 mm$^2$, and 0.1535 mm$^2$ is obtained for the rough ground, hard turned, and fine ground surfaces, respectively. The true contact areas simulated with elasto-plastic material behavior approximate the reference (by elastic simulation with the surfaces in run-in states as input geometry) results (an increase to 22 % and 20 % for the hard-turned and fine-ground surfaces, respectively), see Table 7. As shown in Figure 13b, the simulation time per load cycle is a maximum 30 min for a mesh fineness of four million elements.

The flattening of the roughness peaks that occurs during plasticity leads to a reduction of the maximum contact pressures. Figure 14 shows the elasto-plastic simulated contact pressures after 200 cycles. Compared to the contact pressures with surfaces in the manufactured state from Figure 11, the maximum values are reduced from 15.9 GPa to 4.5 GPa, from 15.9 GPa to 4.3 GPa, and from 17.4 GPa to 4.1 GPa, in each case for the rough ground, hard turned, and fine ground surfaces. For the hard turned and fine ground surfaces, the elasto-plastic simulated maximum contact pressures of 4.3 GPa and 4.1 GPa correlate with the reference (by elastic simulation with the surfaces in run-in states as input geometry) values of 4.3 and 5.7 GPa from the elastic calculation with the measured surface in the run-in state (after $10^7$ revolutions).

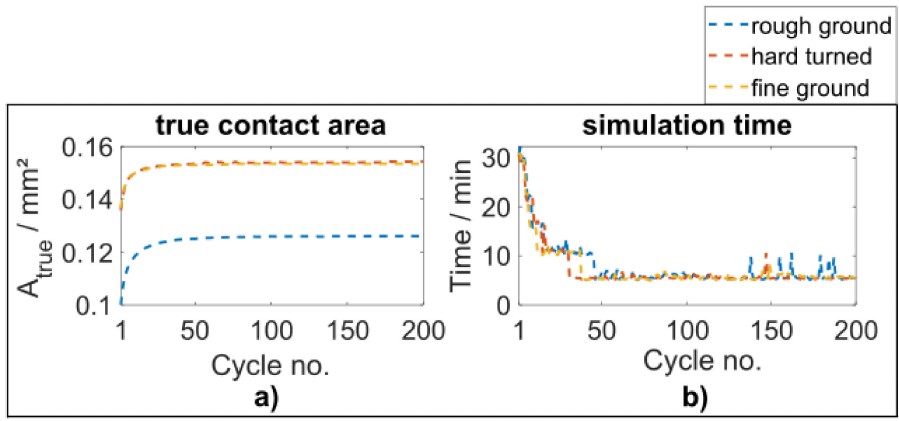

**Figure 13.** (**a**): Elasto-plastic simulated variation of the true contact area under cyclic indentation, (**b**): Simulation time for the respective load cycles.

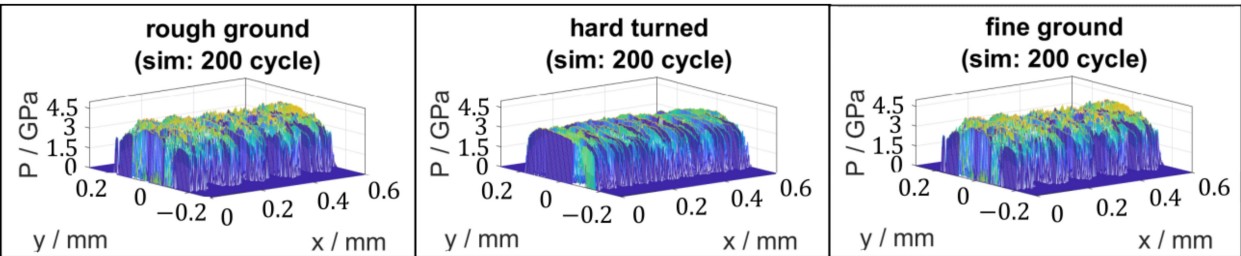

**Figure 14.** Elasto-plastically simulated contact pressure after 200 load cycles.

### 5.2.2. Simulative vs. Experimental Residual Stresses

For the simulative determination of the load-induced residual stresses in the material depth for the different surfaces, an elasto-plastic contact simulation was performed with the previously simulated running surfaces. Figure 15 shows an evaluation of the simulated tangential residual stresses in the rolling direction (ES) in a section through the volume element with the maximum compressive residual stress. The maximum shear stresses also occur on this volume element. Therefore, the fatigue life was related to this volume element. A depth profile through this volume element is also shown in this figure.

As can be seen in the exemplary section in Figure 15, the maximum residual compressive stress values for the hard-turned surface are $-1137$ MPa compared to $-726$ MPa and $-309$ MPa for the rough ground and fine ground surface variants, respectively.

In order to validate the simulated load-induced residual stresses in Figure 15, residual stresses were measured. These measurements were carried out at the Institute of Materials Science and Engineering (WKK) at the RPTU Kaiserslautern-Landau. However, the measured residual stress values for the loaded inner ring specimens contain a superposition of manufacturing-related and load-induced components. To separate the load-induced component approximately (see Figure 16c), residual stress measurements were previously carried out on inner ring samples from the same batch in manufactured state (see Figure 16a). This stress state was subtracted from the stress state on the inner ring after $10^7$ revolutions (see Figure 16b), as shown in Figure 16.

The comparison of the experimentally (see Figure 16c) and simulative (see Figure 15) determined load-induced residual stresses shows that with the half-space model, the range of values of the maximum residual stresses at the surface can be estimated well. The maximum values $-726$ MPa, $-1137$ MPa, and $-309$ MPa were simulatively obtained in comparison to $-678$ MPa, $-972$ MPa, and $-244$ MPa, which were experimentally determined [26], respectively, for the rough ground, hard turned, and fine ground surface.

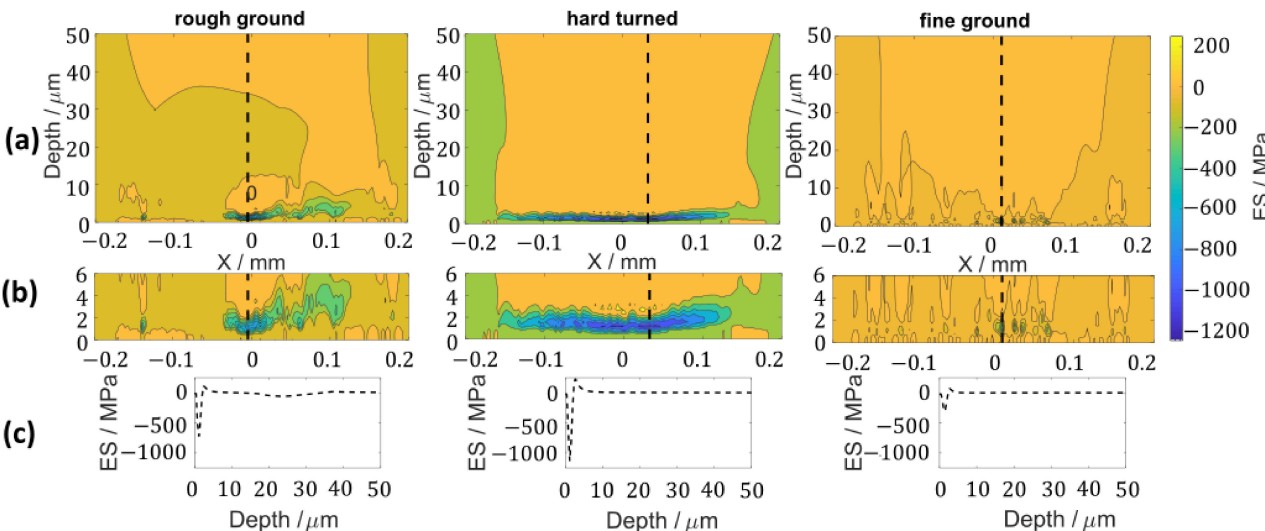

**Figure 15.** Evaluation of the simulated tangential residual stresses on the section plane through the volume element (**a**) a zoom window (**b**) and the maximum compressive residual stress in the section (**c**).

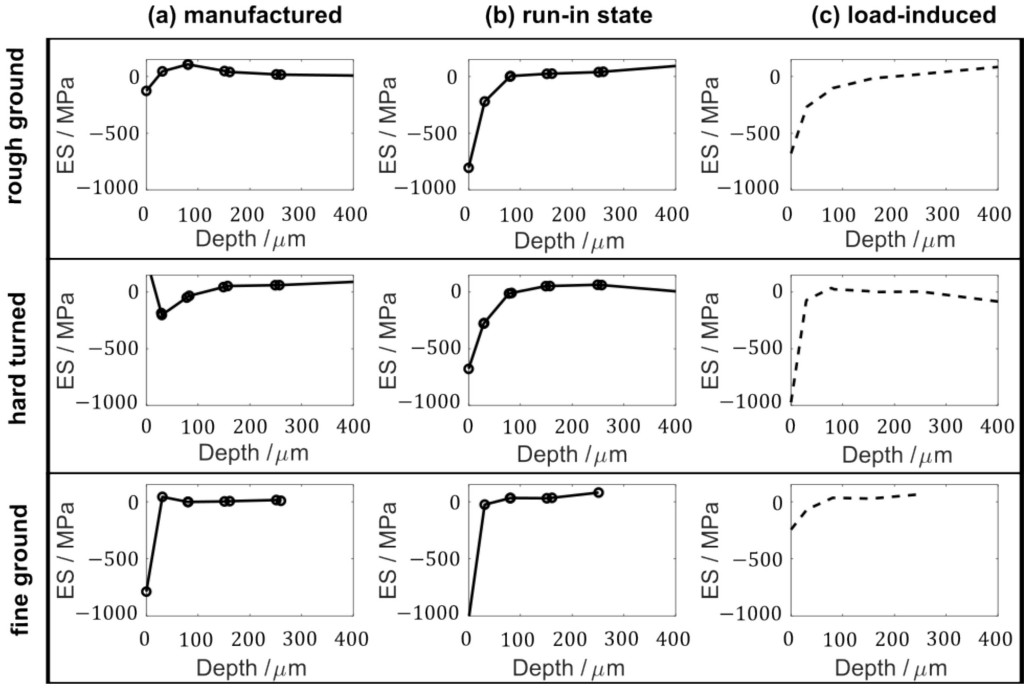

**Figure 16.** Experimental determination of the load-induced residual stress component in (**c**), by subtracting the measured residual stresses on the inner ring in the unloaded state (**a**) from the loaded state (**b**).

### 5.3. Fatigue Life Estimation Using Fatemi-Socie Fatigue Approach

In the Fatemi–Socie fatigue approach, the fatigue life is evaluated locally on a critical volume element. The shear stress hypothesis was used here, and the critical volume element selected was the one with the maximum shear stress. This lies at a depth of 125 μm in the simulation case with the ideally smooth inner ring geometry and a depth of 1.17 μm for all real surfaces. The large difference in the depth of the critical volume element shows that when assuming the ideally smooth surface in the Fatemi–Socie model, classical roller fatigue, which occurs at the material depth, can still be assumed. However, if the real rough inner ring geometry is used, the fatigue starts in the area close to the surface.

For the critical volume element, the components of the stress tensor are determined at each angular position during the passage through the load zone. To reduce computation time, these stress components are computed purely elastically with the previously elasto-plastic computed flattened surface after 200 load cycles, whereby the plasticity is considered.

The validity of this simplified method is verified in the following. For the maximum contact force: 310 N, the components of the stress tensor ($\sigma_{xx} \ldots, \tau_{xz}$), at the critical volume elements in the elasto-plastic simulation case (after 200 cycles) are compared with those from the elastic simulation case with the flattened surface in Table 9.

**Table 9.** Comparison of the elasto-plastic computed stress components at the critical volume element at the 200th load cycle with those from the elastic computation with the elasto-plastic computed flattened surfaces after the 200th load cycle.

|  | **Rough Ground** | | **Hard Turned** | | **Fine Ground** | |
|---|---|---|---|---|---|---|
|  | elastic | elastic-plastic | elastic | elastic-plastic | elastic | elastic-plastic |
| $\sigma_{xx}$/MPa | −1310 | −1379 | −1339 | −1406 | −1312 | 1380 |
| $\sigma_{yy}$/MPa | −2193 | −2319 | −2319 | −2255 | −2117 | −2054 |
| $\sigma_{zz}$/MPa | −3466 | −3477 | −3477 | −3476 | −3331 | −3327 |
| $\tau_{yz}$/MPa | −27 | −27 | −3 | −3 | −23 | −22 |
| $\tau_{xz}$/MPa | −59 | −59 | 48 | 48 | −210 | −210 |
| $\tau_{xz}$/MPa | 8 | 8 | 34 | 34 | −112 | −112 |

From the results in Table 9, similar stress components are obtained for the three surface variants at the critical volume elements for the two methods. It can be concluded that after flattening, elastic contact simulation is sufficient to estimate the stresses.

The stress components at the critical volume elements for the four investigated surfaces are shown in Figure 17.

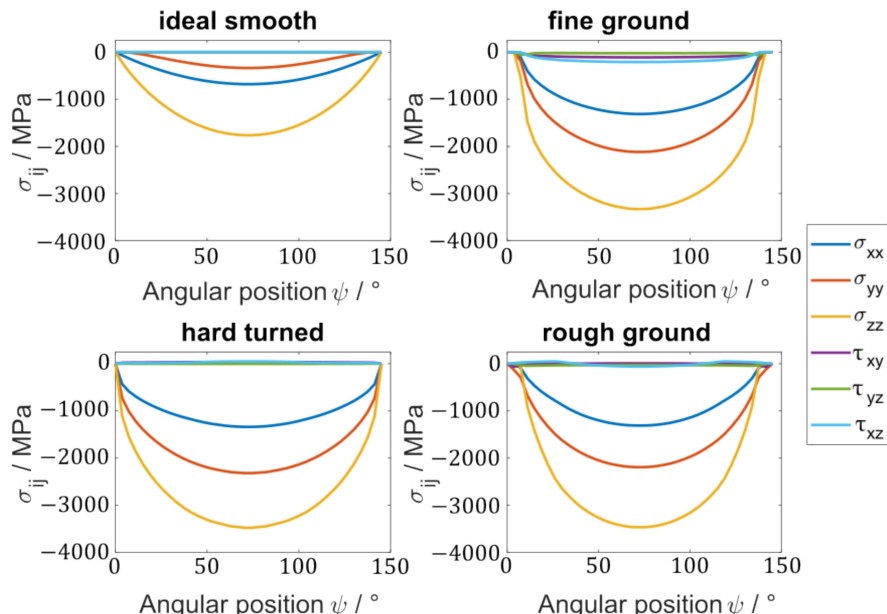

**Figure 17.** Computed stress components on the critical volume element (at depths of 125 μm and 1.17 μm for simulation cases with the ideally smooth and rough inner ring geometry, respectively) for different angular positions.

The equivalent shear strain amplitude $\gamma_{a,eq}$ is calculated as described in Section 3.1 from the two main parameters: shear strain amplitude and maximum normal stress. Figure 18 shows the shear strain amplitudes for all discretized planes $\Pi$ for the four

surface variants ideally smooth, fine ground, hard turned, and rough ground at the critical volume elements. Maximum shear strain amplitudes of 0.0044, 0.0063, 0.0064, and 0.0068 respectively result. Higher shear strain amplitudes lead to high damage parameters in the Fatemi–Socie Equation (1) and thus to lower fatigue life values.

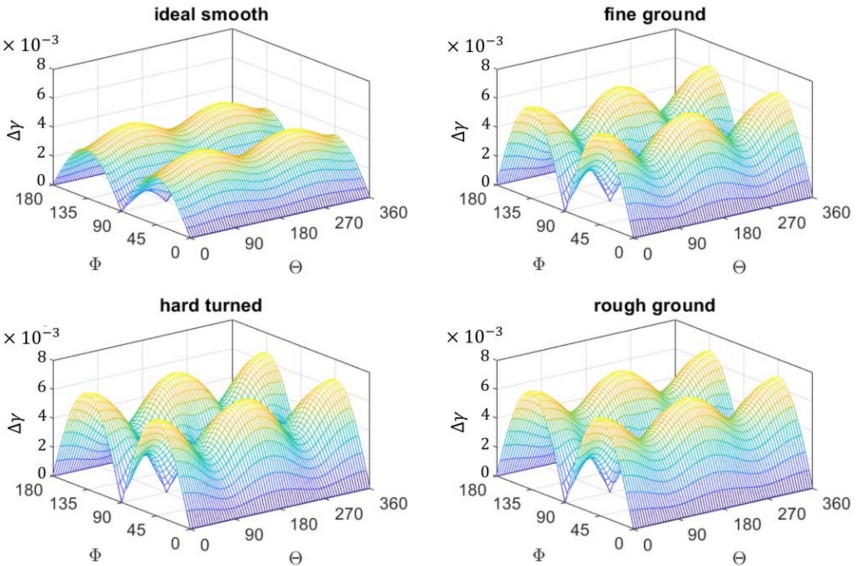

**Figure 18.** Maximum shear strain amplitude on all discretized planes $\Pi$ of the critical volume element for the surface variants: ideally smooth, fine ground, hard turned and rough ground.

It can be observed in Figure 19 that the maximum shear strains occur at the critical planes with the angle combinations $\theta = 90°$, $270°$ and $\phi = 45°$, $135°$ for the ideally smooth surface and $\theta = 0°$, $180°$, $360°$ and $\phi = 45°$, $135°$ for the rough surfaces. The orientation of the critical plane can be determined by the component of the normal vector given in Equation (4).

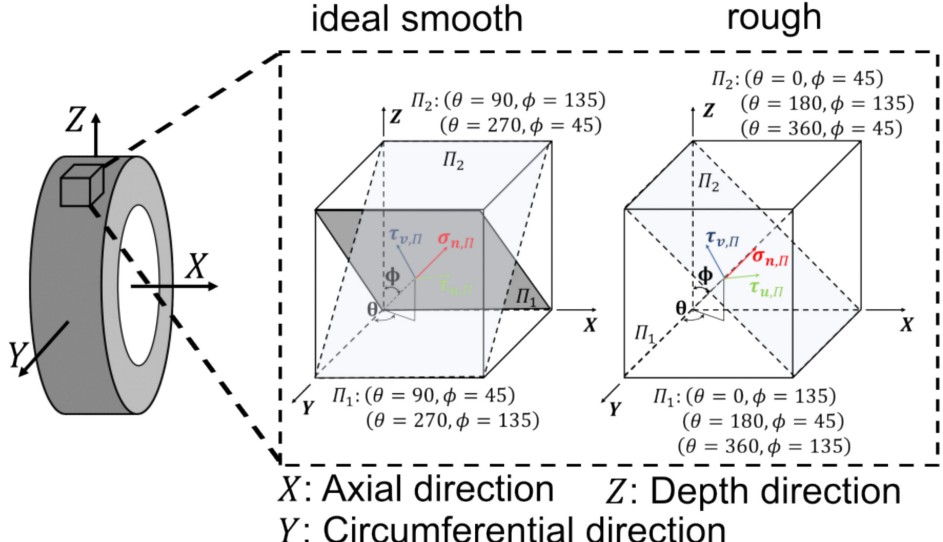

**Figure 19.** Illustration of the orientation of the critical plane in the simulations with ideally smooth and rough inner ring surfaces.

For illustration, the critical planes $\Pi_1$, $\Pi_2$ at the mentioned angle combinations are shown schematically in Figure 19. In the case of ideally smooth surfaces, these lie at an angle of 45° towards the circumferential direction and in the case of rough surfaces also at an angle of 45°, though towards the axial direction.

Figure 20 shows the normal stresses $\sigma_N$ (maximum value marked with the dot) calculated on the critical volume element for the different angular positions. These curves are derived from the stress components in Figure 17 in elastic contact simulations with elasto-plastically calculated flattened surfaces.

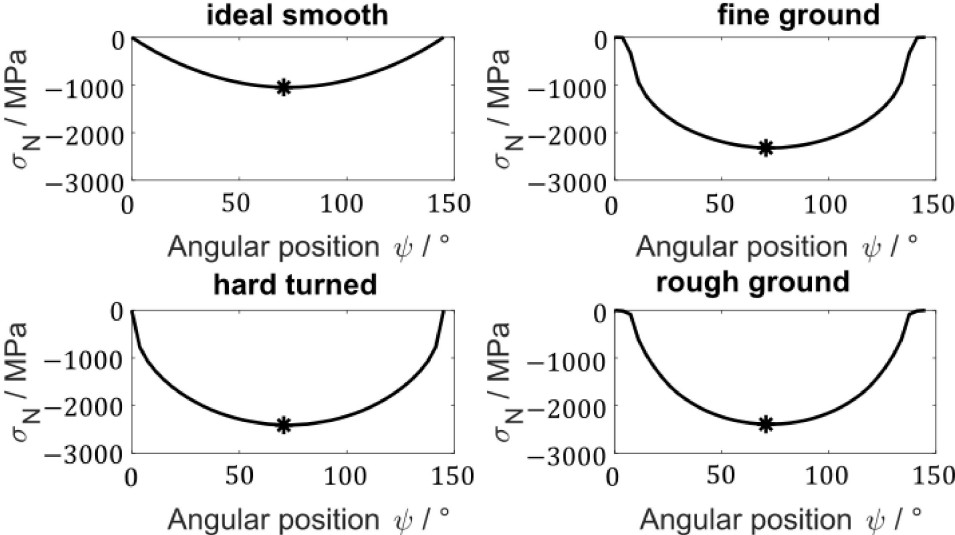

**Figure 20.** Profile of the normal stress perpendicular to the critical plane; maximum pressure normal stress marked with a asterisk.

Table 10 shows the results of the computed damage parameter $\gamma_{a,eq}$, the number of load cycles $N_f$ in the Fatemi–Socie model determined as described in Section 3, and the conversion to the number of inner ring revolutions $N_{IR}$.

**Table 10.** Summary of simulation results: damage parameter $\gamma_{a,eq}$, the critical number of load cycles $N_f$, the number of inner ring revolutions $N_{IR}$, $L_{10}$ calculated according to the ISO standard [1], and experimentally determined fatigue life values life values $B_{10}$.

| Surfaces | $\gamma_{a,eq}$ | $N_f/10^6$ | $N_{IR}/10^6$ | $L_{10}/10^6$ | $B_{10}/10^6$ |
|---|---|---|---|---|---|
| ideally smooth | 0.0027 | 722 | 95 | - | - |
| fine ground | 0.0028 | 381 | 50 | 28.9 | 49.0 |
| hard turned | 0.0028 | 355 | 47 | 27.7 | 78.0 |
| rough ground | 0.0030 | 234 | 30 | 23.7 | 27.1 |

For comparison with the Fatemi–Socie model fatigue life values, the $L_{10}$ (extended modified reference fatigue life) calculated according to [1] and the fatigue life values $B_{10}$ determined experimentally from statistically validated fatigue life tests on four-bearing test rigs (see Section 4.2) with fatigue damage starting from the near surface area are given in Table 10 [26]. In order to calculate the fatigue lifetimes experimentally, the rolling bearing fatigue tests were conducted using the sudden death method, where the tests were continued until the first bearing failed. A total of 24 bearing rings were tested, with six failures observed for each variant. The statistical analysis of the observed fatigue lifetimes was based on the commonly used two-parameter Weibull distribution for rolling bearing lifetime calculations, as described in [32–34].

According to the ISO standard [1], similar fatigue life values result for the three surface variants of rough ground, hard turned, and fine ground. These are significantly smaller in comparison to the experimental results (approx. 41% and 67% differences in each case), especially for the fine-ground and hard-turned surface variants. The local fatigue life model

according to Fatemi–Socie shows higher similarities (with the deviations of 2%, 40 %, and 11%) to the experimental results for the fine ground, hard ground, and rough ground surface variants, respectively, due to the detailed consideration of the surface topography. Compared to the ISO standard, where the roughness is included indirectly via the viscosity ratio $\kappa$, in the presented approach, the surface is considered directly and in much more detail.

## 6. Summary & Conclusions

### 6.1. Summary

A multiscale approach based on two connected contact simulation models (single bearing (MBS) model and single contact (half-space) model) was developed in this paper for the fatigue life estimation of cylindrical roller bearings of the type NU208. The local fatigue approach, according to Fatemi–Socie, was used to determine the critical number of load cycles. For this purpose, the stresses on a defined critical volume element were determined in the half-space model on the microscale. The focus of the investigations was the influence of the surface topography of the inner ring and the resulting plastic behavior on the fatigue life. For this purpose, three manufacturing processes were used, namely rough ground, hard turned, and fine ground with subsequent honing. For each variant, the surface topography and the residual stresses in the manufactured state and in the run-in state (with 2.4 GPa in the rolling element inner ring contact) were first characterized experimentally. The geometry of the rolling elements was assumed to be ideally smooth. The measured values in the run-in state formed the reference values for validating the half-space contact model. From elastic contact simulations, the tribological properties resulting from the different surface topographies have been analyzed. For this purpose, the real contact areas, as well as the stress distribution at the surface topography in run-in state, were determined.

The flattening of the measured surface topography in the loaded state compared to the unloaded state was simulated using elasto-plastic contact simulations. The measured surfaces in the unloaded state were used as the geometric input variable in the contact model. The quasi-static indentation contact between RE and IR with the elasto-plastic material behavior of the inner rings was solved until the convergence of the true contact area.

The stress components at the respective angular positions during the passage of the load zone were determined purely elastically with the elasto-plastically simulated flattened surfaces in order to reduce the computing time.

### 6.2. Conclusions

The hard-turned surface proved to be particularly appropriate from a tribological point of view due to the largest real contact area and the lowest maximum stresses in the loaded state. An approximation of the elasto-plastic simulated real contact area with the surface topography after manufacturing to the elastic contact simulation with consideration of the surface topography in the run-in state (after $10^7$ revolutions) could be observed. A comparison of the simulated residual stress with the experimentally measured values showed that residual stresses can be estimated well with the half-space contact model according to the maximum values. However, the entire range of the residual stress zone was estimated as slightly narrower in the half-space model.

It could be shown that from elasto-plastic simulation in the flattened surface state, similar stress distributions result from pure elastic simulation with the final flattened surface as the input geometry. This finding was used in the fatigue life calculation.

With the local fatigue life model according to Fatemi–Socie, more accurate fatigue life results could be achieved compared to the calculated ISO standard.

## 7. Outlook

This difference observed between experimental and simulated residual stresses could be related to the material parameters used for the Swift hardening model. A more precise

consideration of the depth dependence of the plastic material parameters can be made in the future by means of nanoindentation.

The material parameters used in the right-hand side of the Fatemie-Socie Equation (2), to determine the material strength of the critical volume element were obtained according to the FKM method based only on the experimentally determined tensile strength. In future work, these material parameters could be determined experimentally from the Basquin and Manson–Coffin parameters.

**Author Contributions:** Conceptualization, F.F.F. and L.R.; methodology, F.F.F. and L.R.; software, F.F.F. and L.R.; validation, F.F.F. and L.R.; formal analysis F.F.F. and L.R.; investigation, F.F.F. and L.R.; data curation, F.F.F. and L.R.; writing—original draft preparation, F.F.F. and L.R.; visualization, F.F.F. supervision, O.K. and B.S.; project administration, B.S.; funding acquisition, B.S.; All authors have read and agreed to the published version of the manuscript.

**Funding:** This research was funded by German Research Foundation (DFG, Deutsche Forschungsgemeinschaft), SFB 926–projectnumber: 172116086.

**Data Availability Statement:** Data are available on request from the authors.

**Acknowledgments:** The presented work was funded by the German Research Foundation (DFG, Deutsche Forschungsgemeinschaft) as a transfer project within the collaborative research center SFB 926–projectnumber: 172116086, which is gratefully acknowledged. The authors would also like to thank the company DIGITAL SURF for providing the software MountainsMap.

**Conflicts of Interest:** The authors declare no conflict of interest.

## Abbreviations

**General abbrevations**

| | |
|---|---|
| $D_R$; $D_{IR}$ | Roller diameter; Inner ring diameter |
| FE | Finite Element |
| IR | Inner ring |
| $L_{eff}$ | Effective contact length |
| $L_x$; $L_y$ | Length of the computational domain in axial direction (x); and circumferential direction (y) |
| MBS | Multibody simulation |
| OR | Outer ring |
| RE | Rolling element |
| $V_{IR}$; $V_{\text{critical}}$ | Discretised inner ring volume area (calculation volume); critical volume element in discretised inner ring volume area |
| $\psi$; $\psi_E$, $\psi_A$ | Angular position in the load zone; on entry, and exit of the load zone |
| $\sigma_v$; $\sigma_{F0}$ | Equivalent stress; start yield stress |

**Material parameters (elasto-plastic)**

| | |
|---|---|
| B; C; n | Plastic material parameters |
| $E$; $\nu$ | Elastic material parameters |

**Material parameters (Fatemi–Socie Model)**

| | |
|---|---|
| $b_y$ | Fatigue exponent for pure torsional loading |
| $c_y$ | Ductility exponent for pure torsional loading |
| $G$ | Shear modulus |
| $k_e$ | Nondimensional material constant (normal stress sensitivity coefficient) |
| $N_f$ | Critical number of load cycles |
| $\gamma_{a,eq}$ | Equivalent shear strain amplitude |
| $\gamma_{max}$ | Maximum shear strain amplitude in the critical plane |
| $\gamma_f'$ | Cyclic ductility coefficient for pure torsional loading |
| $\tau_f'$ | Cyclic fatigue coefficient for pure torsional loading |
| $\sigma_F$ | Yield stress |
| $\sigma_{n,max}$ | Max. normal stress perpendicular to the critical plane |

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
