# Peer review of "Study of the Plastic Behavior of Rough Bearing Surfaces Using a Half-Space Contact Model and the Fatigue Life Estimation According to the Fatemi–Socie Model"

_lubricants, doi:10.3390/lubricants11030133_

Round 1

Reviewer 1 Report

Dear Author(s), the manuscript ‘Study of the plastic behavior of rough bearing surfaces using a half-space contact model and the fatigue life estimation according to the Fatemi-Socie model’, Manuscript ID: lubricants-2221562, have some weakness that must be revised properly.

Below are listed some of the most significant comments:

1.      In the ‘Introduction’ section more words on work motivation should be placed not only those in lines 46-51.

2.      According to the previous comment, there is no critical review in the first section. From that matter, the motivation of work including highlighting the lack in the current state of knowledge is still hidden.

3.      Values of the variables of contact models, included in section 2.1, especially in Table 1, should be justified. Currently, it looks selected arbitrarily.

4.      The novelty must be highlighted and separated from the solutions already known, especially in sections 2 and 3.

5.      Similarly to the previous suggestion, equations (4) and (5) must be referenced, if not newly proposed by the Author(s).

6.      In section 4.1, details of the surface roughness measurement are not provided, respectively, only a technique (confocal) is presented. In fact, in the studied case, the main issue is not related to the measurement but to comparing modelled data with those measured, nevertheless, the accuracy of the measurement can affect both the final responses (conclusions) and results.

7.      According to the previous comment, the accuracy of the roughness measurement should be even mentioned. The Author(s) must refer to the measurement uncertainty (repeatability) and measurement noise (errors), e.g.:

(1)   https://doi.org/10.1088/2051-672X/3/3/035004

(2)   https://doi.org/10.3390/s22030791

(3)   https://doi.org/10.1016/j.cirp.2014.03.086

Without consideration of the measurement precision of the final results, respectively validation is lost.

8.      More details on the machining (rough ground, hard turned and fine ground) must be provided. When presenting any advice to the reader, the type of surface must be precise.

9.      It is not clear in section 4.2.1, what were the main criteria for the roughness comparison for simulated and measured data. Why is it even provided? Is it essential? In lines 418-419, it was mentioned that the ‘detailed consideration of the surface topography’ was presented, but where? Comparison of some random profiles?

10.  The ‘Conclusion’ section must be improved. Firstly, must be divided into separate, numbered gaps. Secondly, the citations (lines 452 and 463) are not pleasured to be found in this section. It can be received that novelty was not appropriately highlighted, as mentioned in one of the previous comments. Finally, I suggest providing an outlook as a separate section. Currently, this section (5) is too long and makes the reader lost and difficult to follow what the Author(s) are trying to convey.

Moreover, some additional (editorial) modifications are required as well:

11.  The affiliation should not be numbered and there is only one (line 6).

12.  In line 38, surnames should not be in capital letters, if not citing the method named by the authors' surnames.

13.  The ‘References’ should be provided with full DOI links, if exist.

14.  The quality of some figures, e.g. Figure 4 and Figure 11, must be improved.

From the above, the reviewed manuscript must be improved significantly before any further processing of the Lubricants journal, if allowed by the Editor.

Author Response

Dear reviewer,

in the attached PDF- file a reply to your comments has been given and the manuscript has been revised.

Yours sincerely

Reviewer 2 Report

Review on the Manuscript:

Paper ID: lubricants-2221562

Title: Study of the plastic behavior of rough bearing surfaces using a half-space contact model and the fatigue life estimation according to the Fatemi-Socie model

A multi-scale approach based on two connected contact simulation models was developed in this paper for the fatigue life estimation of cylindrical roller bearings of the type NU208. The local fatigue approach according to Fatemi-Socie was used to determine the critical number of load cycles.

The contents are relevant to this journal. I recommend its publication but before publication, I suggest following revision:

1.      The novelty of the work should be highlighted to real physics phenomena in the introduction.

2.      The authors should try to give advantageous of using of their method compared to others.

3.      The obtained findings of this work should be compared to experimental results or at-least with other published results in the literature.

4.      After the numbered equations, put a comma if the following sentence starts with a lowercase letter, respectively put a point if the following sentence starts with a capital letter.

5.      The results presented in the graphs are not easy to understand. The figures are too cramped, the authors should find a clearer form of presentation.

 If the authors take into account all these corrections, then this manuscript deserves to be published

Author Response

(The authors gave the same response as above.)

Reviewer 3 Report

This is a good work. The authors studied the contact behavior of the inner ring in a rolling-element bearing using a half-space contact model. The cycling load was simulated using Simpack (?); its peak load then was used as applied load to the micro contact area (Lx times Ly) with a scaled factor of Ly/L_eff. With this multiscale model, the authors simulated the rough inner-ring surface problems to investigate the real contact areas, roughness profiles, contact pressure distributions, residual stresses, stress components on critical elements, and fatigue life estimation using Fatemi-Socie model. The profiles of surface roughness were measured from the real situation: rough ground, hard turned and fine ground before and after running-in process. The simulation results were validated with the experiment. For the fatigue life estimations, they were compared with the ISO standard and fatigue life tests.

In the reviewer’s opinion, the most crucial part of the proposed model is the half-space model: it accurately predicted how the asperities stressed and deformed elastically and elasto-plastically. The authors need to spend more paragraphs to discuss this micro model.

Some mistakes need to be fixed are as follows:

1.     References [11] and [24] had never been cited.

2.     In Figure 10, the columns one and two seem to be mislabeled. The 1st column seems to be the manufactured and the 2nd the run-in state.

3.     Lines 287 to 288: what’s the reference results.

4.     There are format inconsistencies in the reference list.

Author Response

(The authors gave the same response as above.)

Round 2

Reviewer 1 Report

Dear author(s), the manuscript titled ‘Study of the plastic behavior of rough bearing surfaces using a half-space contact model and the fatigue life estimation according to the Fatemi-Socie model’, Manuscript ID: lubricants-2221562,, has been improved suitably so, respectively, can be further processed by the Lubricants journal.

Thank you for your full responses that, in their current form, were addressed properly and make the manuscript appropriate for publication in a quality journal as the Lubricants is.